# Optimal Tagging with Markov Chain Optimization

**Nir Rosenfeld**
School of Computer Science and Engineering
Hebrew University of Jerusalem
nir.rosenfeld@mail.huji.ac.il

**Amir Globerson**
The Blavatnik School of Computer Science
Tel Aviv University
gamir@post.tau.ac.il

## Abstract

Many information systems use tags and keywords to describe and annotate content. These allow for efficient organization and categorization of items, as well as facilitate relevant search queries. As such, the selected set of tags for an item can have a considerable effect on the volume of traffic that eventually reaches an item.

In tagging systems where tags are exclusively chosen by an item's owner, who in turn is interested in maximizing traffic, a principled approach for assigning tags can prove valuable. In this paper we introduce the problem of optimal tagging, where the task is to choose a subset of tags for a new item such that the probability of browsing users reaching that item is maximized.

We formulate the problem by modeling traffic using a Markov chain, and asking how transitions in this chain should be modified to maximize traffic into a certain state of interest. The resulting optimization problem involves maximizing a certain function over subsets, under a cardinality constraint.

We show that the optimization problem is NP-hard, but has a $(1-\frac{1}{e})$-approximation via a simple greedy algorithm due to monotonicity and submodularity. Furthermore, the structure of the problem allows for an efficient computation of the greedy step. To demonstrate the effectiveness of our method, we perform experiments on three tagging datasets, and show that the greedy algorithm outperforms other baselines.

## 1 Introduction

To allow for efficient navigation and search, modern information systems rely on the usage of non-hierarchical tags, keywords, or labels to describe items and content. These tags are then used either explicitly by users when searching for content, or implicitly by the system to recommend related items or to augment search results.

Many online systems where users can create or upload content support tagging. Examples of such systems are media-sharing platforms, social bookmarking websites, and consumer to consumer auctioning services. While in some systems any user can tag any item, in many ad-hoc systems tags are exclusively set by the item's owner alone. She, in turn, is free to select any set of tags or keywords which she believes best describe the item. Typically, the only concrete limitation is on the number of tags, words, or characters used. Tags are often chosen on a basis of their ability to best describe, classify, or categorize items and content. By choosing relevant tags, users aid in creating a more organized information system. However, content owners may have their own individual objective, such as maximizing the exposure of their items to other browsing users. This is true for many artists, artisans, content creators, and merchants whose services and items are provided online.

This suggests that choosing tags should in fact be done strategically. For instance, for a user uploading a new song, tagging it as 'Rock' may be informative, but will probably only contribute marginally to the song's traffic, as the competition for popularity under this tag can be fierce. On the other hand, choosing a unique or obscure tag may be appealing, but will not help much either. Strategic tagging

or keyword selection is clearly exhibited in search systems, where keywords are explicitly used for filtering and ordering search results or ad placements, and users have a clear incentive of maximizing an item's exposure. Nonetheless, the selection process is typically heuristic.

Recent years have seen an abundance of work on methods for user-specific tag recommendations [8, 10, 5]. Such methods aim to support collaborative tagging systems, where any user can tag any item in the repository. In contrast, in this paper we take a complementary perspective and focus on *taxonomic* tagging systems, where only the owner of an item can determine its tags. We formalize the task of *optimal tagging* and suggest an efficient, provably-approximate algorithm. While the problem is shown to be NP-hard, we prove that the objective is in fact monotone and submodular, which suggests a straightforward greedy $(1 - \frac{1}{e})$-approximation algorithm [13]. We also show how the greedy step, which consists of solving a set of linear equations, can be greatly simplified. This results in a significant improvement of runtime.

We begin by modeling a user browsing a tagged information system as a random walk. Items and tags act as states in a Markov chain, whose transition probabilities describe the probability of users transitioning between items and tags. Given a new item, our task is to choose a subset of $k$ tags for this item. When an item is tagged, positive probabilities are assigned to transitioning both from the tag to the item and from the item to the tag. Our objective is to choose the subset of $k$ tags which will maximize traffic to that item, namely the probability of a random walk reaching the item. Intuitively, tagging an item causes probability to flow from the tag to the item, on account of other items with this tag. Our goal is to take as much probability mass as possible from the system as a whole.

Our method shares some similarities with other PageRank (PR, [2]) based methods, which optimize measures based on the stationary distribution [14, 4, 6, 15]. Here we argue that our approach, which focuses on maximizing the probability of a random walk reaching a new item's state, is better suited to the task of optimal tagging. First, an item's popularity should only increase when assigned a new tag. Since tagging an item creates bidirectional links, its stationary probability may undesirably decrease. Hence, optimizing the PR of an item will lead to an undesired non-monotone objective [1]. Second, PR considers a single Markov chain for all items, and is thus not item-centric. In contrast, our method considers a unique instance of the transition system for every item we consider. While an item-specific Personalized PR based objective can be constructed, it would consider random walks *from* a given state, not to it. Third, a stationary distribution does not always exist, and hence may require modifications of the Markov chain. Finally, optimizing PR is known to be hard. While some approximations exist, our method provides superior guarantees and potentially better runtime [16].

Although the Markov chain model we propose for optimal tagging is bipartite, our results apply to general Markov chains. We therefore first formulate a general problem in Sec. 3, where the task is to choose $k$ states to link a new state to such that the probability of reaching that state is maximal. Then, in Sec. 4 we prove that this problem is NP-hard by a reduction from vertex cover. In Sec. 5 we prove that for a general Markov chain the optimal objective is both monotonically non-decreasing and submodular. Based on this, in Sec. 6 we suggest a basic greedy $(1 - \frac{1}{e})$-approximation algorithm, and describe a method for significantly improving its runtime. In Sec. 7 we revisit the optimal tagging problem and show how to construct a bipartite Markov chain for a given tag-supporting information system. In Sec. 8 we present experimental results on three tagging datasets (musical artists in Last.fm, bookmarks in Delicious, and movies in Movielens) and show that our algorithm outperforms other baselines. Concluding remarks are given in Sec. 9.

## 2   Related Work

One of the main roles of tags is to aid in the categorization and classification of content. An active line of research in tagging systems focuses on the task of tag recommendations, where the goal is to predict the tags a given user may assign an item. This task is applicable in collaborative tagging systems and folksonomies, where any user can tag any item. Methods for this task are based on random walks [8, 10] and tensor factorization [5]. While the goal in tag recommendation is also to output a set of tags, our task is very different in nature. Tag recommendation is a prediction task for item-user pairs, is based on ground-truth evaluation, and is applied in collaborative tagging systems. In contrast, ours is an item-centric optimization task for tag-based taxonomies, and is counterfactual in nature, as the selection of tags is assumed to affect future outcomes.

A line of work similar to ours is optimizing the PageRank of web pages in different settings. In [4] the authors consider the problem of computing the maximal and minimal PageRank value for a set of "fragile" links. The authors of [1] analyze the effects of additional outgoing links on the PageRank value. Perhaps the works most closely related to ours are [16, 14], where an approximation algorithm is given for the problem of maximizing the PageRank value by adding at most $k$ incoming links. The authors prove that the probability of reaching a web page is submodular and monotone in a fashion similar to ours (but with a different parameterization), and use it as a proxy for PageRank.

Our framework uses absorbing Markov chains, whose relation to submodular optimization has been explored in [6] for opinion maximization and in [12] for computing centrality measures in networks. Following the classic work of Nemhauser [13], submodular optimization is now a very active line of research. Many interesting optimization problems across diverse domains have been shown to be submodular. Examples include sensor placement [11] and social influence maximization [9].

## 3  Problem Formulation

Before presenting our approach to optimal tagging, we first describe a general combinatorial optimization task over Markov chains, of which optimal tagging is a special case. Consider a Markov chain over $n + 1$ states. Assume there is a state $\sigma$ to which we would like to add $k$ new incoming transitions, where w.l.o.g. $\sigma = n + 1$. In the tagging problem, $\sigma$ will be an item (e.g., song or product) and the incoming transitions will be from possible tags for the item, or from related items.

The optimization problem is then to choose a subset $S \subseteq [n]$ of $k$ states so as to maximize the probability of visiting $\sigma$ at some point in time. Formally, let $X_t \in [n + 1]$ be the random variable corresponding to the state of the Markov chain at time $t$. Then the optimal tagging problem is:

$$\max_{S \subseteq [n], |S| \leq k} \mathbb{P}_S \left[ X_t = \sigma \text{ for some } t \geq 0 \right] \tag{1}$$

At first glance, it is not clear how to compute the objective function in Eq. (1). However, with a slight modification of the Markov chain, the objective function can be expressed as a simple function of the Markov chain parameters, as explained next.

In general, $\sigma$ may have outgoing edges, and random walks reaching $\sigma$ may continue to other states afterward. Nonetheless, as we are only interested in the probability of *reaching* $\sigma$, the states visited after $\sigma$ have no effect on our objective. Hence, the edges out of $\sigma$ can be safely replaced with a single self-edge without affecting the probability of reaching $\sigma$. This essentially makes $\sigma$ an *absorbing state*, and our task becomes to maximize the probability of the Markov chain being absorbed in $\sigma$. In the remainder of the paper we consider this equivalent formulation.

When the Markov chain includes other absorbing states, optimizing over $S$ can be intuitively thought of as trying to transfer as much probability mass from the competing absorbing states to $\sigma$, under a budget on the number of states that can be connected to $\sigma$.[1] As we discuss in Section 7, having competing absorbing states arises naturally in optimal tagging.

To fully specify the problem, we need the Markov chain parameters. Denote the initial distribution by $\boldsymbol{\pi}$. For the transition probabilities, each node $i$ will have two sets of transitions: one when it is allowed to transition to $\sigma$ (i.e., $i \in S$) and one when no transition is allowed. Using two distinct sets is necessary since in both cases outgoing probabilities must sum to one. We use $q_{ij}$ to denote the transition probability from state $i$ to $j$ when transition from $i$ to $\sigma$ is not allowed, and $q_{ij}^+$ when it is. We also denote the corresponding transition matrices by $Q$ and $Q^+$.

It is natural to assume that when adding a link from $i$ to $\sigma$, transition into $\sigma$ will become more likely, and transition to other states can only be less likely. Thus, we add the assumptions that:

$$\forall i \; : \; 0 = q_{i\sigma} \leq q_{i\sigma}^+, \qquad \forall i \,, \; \forall j \neq \sigma \; : \; q_{ij}^+ \leq q_{ij} \tag{2}$$

Given a subset $S$ of states from which transitions to $\sigma$ are allowed, we construct a new transition matrix, taking corresponding rows from $Q$ and $Q^+$. We denote this matrix by $\rho(S)$, with

$$\rho_{ij}(S) = \left\{ \begin{array}{ll} q_{ij}^+ & i \in S \\ q_{ij} & i \notin S \end{array} \right. \tag{3}$$

# 4 NP-Hardness

We now show that for a general Markov chain, the optimal tagging problem in Eq. (1) is NP-hard by a reduction from vertex cover. Given an undirected graph $G = (V, E)$ with $n$ nodes as input to the vertex cover problem, we construct an instance of optimal tagging such that there exists a vertex cover $S \subseteq V$ of size at most $k$ iff the probability of reaching $\sigma$ reaches some threshold.

To construct the absorbing Markov chain, we create a transient state $i$ for every node $i \in V$, and add two absorbing states $\varnothing$ and $\sigma$. We set the initial distribution to be uniform, and for some $0 < \epsilon < 1$ set the transitions for transient states $i$ as follows:

$$q_{ij}^+ = \left\{ \begin{array}{ll} 1 & j = \sigma \\ 0 & j \neq \sigma \end{array} \right. , \qquad q_{ij} = \left\{ \begin{array}{ll} 0 & j = \sigma \\ \epsilon & j = \varnothing \\ \frac{1-\epsilon}{deg(i)} & \text{otherwise} \end{array} \right. \tag{4}$$

Let $U \subseteq V$ of size $k$, and $S(U)$ the set of states corresponding to the nodes in $U$. We claim that $U$ is a vertex cover in $G$ iff the probability of reaching $\sigma$ when $S(U)$ is chosen is $1 - \frac{(n-k)}{n}\epsilon$.

Assume $U$ is a vertex cover. For every $i \in S(U)$, a walk starting in $i$ will reach $\sigma$ with probability 1 in one step. For every $i \notin S(U)$, with probability $\epsilon$ a walk will reach $\varnothing$ in one step, and with probability $1 - \epsilon$ it will visit one of its neighbors $j$. Since $U$ is a vertex cover, it will then reach $\sigma$ in one step with probability 1. Hence, in total it will reach $\sigma$ with probability $1 - \epsilon$. Overall, the probability of reaching $\sigma$ is $\frac{k+(n-k)(1-\epsilon)}{n} = 1 - \frac{(n-k)}{n}\epsilon$ as needed. Note that this is the maximal possible probability of reaching $\sigma$ for *any* subset of $V$ of size $k$.

Assume now that $U$ is not a vertex cover, then there exists an edge $(i, j) \in E$ such that both $i \notin S(U)$ and $j \notin S(U)$. A walk starting in $i$ will reach $\varnothing$ in one step with probability $\epsilon$, and in two steps (via $j$) with probability $\epsilon \cdot q_{ij} > 0$. Hence, it will reach $\sigma$ with probability strictly smaller than $1 - \epsilon$, and the overall probability of reaching $\epsilon$ will be strictly smaller than $1 - \frac{(n-k)}{n}\epsilon$.

# 5 Proof of Monotonicity and Submodularity

Denote by $\mathbb{P}_S [A]$ the probability of event $A$ when transitions from $S$ to $\sigma$ are allowed. We define:

$$c_i^{(k)}(S) = \mathbb{P}_S [X_t = \sigma \text{ for some } t \leq k | X_0 = i] \tag{5}$$

$$c_i(S) = \mathbb{P}_S [X_t = \sigma \text{ for some } t | X_0 = i] = \lim_{k \to \infty} c_i^{(k)} \tag{6}$$

Using $\boldsymbol{c}(S) = (c_1(S), \ldots, c_n(S))$, the objective in Eq. (1) now becomes:

$$\max_{S \subseteq [n], |S| \leq k} f(S), \qquad f(S) = \langle \boldsymbol{\pi}, \boldsymbol{c}(S) \rangle = \mathbb{P}_S [X_t = \sigma \text{ for some } t] \tag{7}$$

We now prove that $f(S)$ is both monotonically non-decreasing and submodular.

## 5.1 Monotonicity

When a link is created from $i$ to $\sigma$, the probability of reaching $\sigma$ directly from $i$ increases. However, due to the renormalization constraints, the probability of reaching $\sigma$ via longer paths may decrease. Trying to prove that for every random walk $f$ is monotone and using additive closure is bound to fail. Nonetheless, our proof of monotonicity shows that the overall probability cannot decrease.

**Theorem 5.1.** *For every $k \geq 0$ and $i \in [n]$, $c_i^{(k)}$ is non-decreasing. Namely, for all $S \subseteq [n]$ and $z \in [n] \setminus S$, it holds that $c_i^{(k)}(S) \leq c_i^{(k)}(S \cup \{z\})$.*

*Proof.* We prove by induction on $k$. For $k = 0$, as $\boldsymbol{\pi}$ is independent of $S$ and $z$, we have:

$$c_i^0(S) = \boldsymbol{\pi}_\sigma \mathbb{1}_{\{i=\sigma\}} = c_i^0(S \cup \{z\})$$

Assume now that the claim holds for some $k \geq 0$. For any $T \subseteq [n]$, it holds that:

$$c_i^{(k+1)}(T) = \sum_{j=1}^{n} \rho_{ij}(T) c_j^{(k)}(T) + \rho_{i\sigma} \mathbb{1}_{\{i \in T\}} \tag{8}$$

We separate into cases. When $i \neq z$, we have:

$$i \in S: \quad c_i^{(k+1)}(S) = \sum_{j=1}^n q_{ij}^+ c_j^{(k)}(S) + q_{i\sigma}^+ \leq \sum_{j=1}^n q_{ij}^+ c_j^{(k)}(S \cup z) + q_{i\sigma}^+ = c_i^{(k+1)}(S \cup z) \quad (9)$$

$$i \notin S: \quad c_i^{(k+1)}(S) = \sum_{j=1}^n q_{ij} c_j^{(k)}(S) \leq \sum_{j=1}^n q_{ij} c_j^{(k)}(S \cup z) = c_i^{(k+1)}(S \cup z) \quad (10)$$

using the inductive assumption and Eq. (8). When $i = z$, we have:

$$
\begin{aligned}
c_i^{(k+1)}(S) &\leq \sum_{j=1}^n q_{ij} c_j^{(k)}(S \cup z) = \sum_{j=1}^n q_{ij}^+ c_j^{(k)}(S \cup z) + \sum_{j=1}^n (q_{ij} - q_{ij}^+) c_j^{(k)}(S \cup z) \\
&\leq \sum_{j=1}^n q_{ij}^+ c_j^{(k)}(S \cup z) + \sum_{j=1}^n (q_{ij} - q_{ij}^+) = \sum_{j=1}^n q_{ij}^+ c_j^{(k)}(S \cup z) + q_{z\sigma}^+ = c_i^{(k+1)}(S \cup z)
\end{aligned}
$$

due to to $q_{ij} \geq q_{ij}^+$, $c \leq 1$, $\sum_{j=1}^n q_{ij} = 1$, and $\sum_{j=1}^n q_{ij}^+ = 1 - q_{i\sigma}^+$. $\qquad \square$

**Corollary 5.2.** $\forall i \in [n]$, $c_i(S)$ *is non-decreasing, hence* $f(S) = \langle \boldsymbol{\pi}, \boldsymbol{c}(S) \rangle$ *is non-decreasing.*

## 5.2 Submodularity

Submodularity captures the principle of diminishing returns. A function $f(S)$ is submodular if:

$$\forall X \subseteq Y \subseteq [n], \ z \notin X, \quad f(X \cup \{z\}) - f(X) \geq f(Y \cup \{z\}) - f(Y)$$

In what follows we will use the following equivalent definition:

$$\forall S \subseteq [n], \ z_1, z_2 \in [n] \setminus S, \quad f(S \cup \{z_1\}) + f(S \cup \{z_2\}) \geq f(S \cup \{z_1, z_2\}) + f(S) \quad (11)$$

Using this formulation, we now show that $f(S)$ as defined in Eq. (7) is submodular.

**Theorem 5.3.** *For every* $k \geq 0$ *and* $i \in [n]$, $c_i^{(k)}(S)$ *is a submodular function.*

*Proof.* We prove by induction on $k$. For $k = 0$, once again $\boldsymbol{\pi}$ is independent of $S$ and hence $c_i^0$ is modular. Assume now that the claim holds for some $k \geq 0$. For brevity we define:

$$c_i^{(k)} = c_i^{(k)}(S), \quad c_{i,1}^{(k)} = c_i^{(k)}(S \cup \{z_1\}), \quad c_{i,2}^{(k)} = c_i^{(k)}(S \cup \{z_2\}), \quad c_{i,12}^{(k)} = c_i^{(k)}(S \cup \{z_1, z_2\})$$

We'd like to show that $c_{i,1}^{(k+1)} + c_{i,2}^{(k+1)} \geq c_{i,12}^{(k+1)} + c_i^{(k+1)}$. For every $j \in [n]$, we'll prove that:

$$\rho_{ij}(S \cup \{z_1\}) c_{j,1}^{(k)} + \rho_{ij}(S \cup \{z_2\}) c_{j,2}^{(k)} \geq \rho_{ij}(S \cup \{z_1, z_2\}) c_{j,12}^{(k)} + \rho_{ij}(S) c_j^{(k)} \quad (12)$$

By summing over all $j \in [n]$ and adding $\rho_{i\sigma} \mathbb{1}_{\{i \in T\}}$ we get Eq. (8) and conclude our proof.

We separate into different cases for $i$. If $i \in S$, then we have $\rho_{ij}(S \cup \{z_1, z_2\}) = \rho_{ij}(S \cup \{z_1\}) = \rho_{ij}(S \cup \{z_2\}) = \rho_{ij}(S) = q_{ij}^+$. Similarly, if $i \notin S \cup \{z_1, z_2\}$, then all terms now equal $q_{ij}$. Eq. (12) then follows from the inductive assumption.

Assume $i = z_1$ (and analogously for $i = z_2$). From the assumption in Eq. (2) we can write $q_{ij} = (1 + \alpha) q_{ij}^+$ for some $\alpha \geq 0$. Then Eq. (12) becomes:

$$q_{ij}^+ c_{j,1}^{(k)} + (1 + \alpha) q_{ij}^+ c_{j,2}^{(k)} \geq q_{ij}^+ c_{j,12}^{(k)} + (1 + \alpha) q_{ij}^+ c_j^{(k)} \quad (13)$$

Divide by $q_{ij}^+ > 0$ if needed and reorder to get:

$$\left( c_{j,1}^{(k)} + c_{j,2}^k - c_{j,12}^{(k)} - c_j^{(k)} \right) + \alpha (c_{j,2}^k - c_j^{(k)}) \geq 0 \quad (14)$$

This indeed holds since the first term is non-negative from the inductive assumption, and the second term is non-negative because of monotonicity and $\alpha \geq 0$. $\qquad \square$

**Corollary 5.4.** $\forall i \in [n]$, $c_i(S)$ *is submodular, hence* $f(S) = \langle \boldsymbol{\pi}, \boldsymbol{c}(S) \rangle$ *is submodular.*

**Algorithm 1**

---

1: **function** SIMPLEGREEDYTAGOPT($Q, Q^+, \boldsymbol{\pi}, k$)      $\triangleright$ See supp. for efficient implementation
2:      Initialize $S = \emptyset$
3:      **for** $i \leftarrow 1$ to $k$ **do**
4:          **for** $z \in [n] \setminus S$ **do**
5:              $\boldsymbol{c} = (I - A(S \cup \{z\})) \setminus \boldsymbol{b}(S \cup \{z\})$      $\triangleright$ $A, \boldsymbol{b}$ are set by $Q, Q^+$ using Eqs. (3), (15)
6:              $v(z) = \langle \boldsymbol{\pi}, \boldsymbol{c} \rangle$
7:          $S \leftarrow S \cup \mathrm{argmax}_z \, v(z)$
8:      Return $S$

---

## 6  Optimization

Maximizing submodular functions is hard in general. However, a classic result by Nemhauser [13] shows that a non-decreasing submodular set function, such as our $f(S)$, can be efficiently optimized via a simple greedy algorithm, with a guaranteed $(1 - \frac{1}{e})$-approximation of the optimum. The greedy algorithm initializes $S = \emptyset$, and then sequentially adds elements to $S$. For a given $S$, the algorithm iterates over all $z \in [n] \setminus S$ and computes $f(S \cup \{z\})$. Then, it adds the highest scoring $z$ to $S$, and continues to the next step. We now discuss its implementation for our problem.

Computing $f(S)$ for a given $S$ reduces to solving a set of linear equations. For transient states $\{1, \ldots, n - r\}$ and absorbing states $\{n - r + 1, \ldots, n + 1 = \sigma\}$, the transition matrix $\rho(S)$ becomes:

$$\rho(S) = \begin{pmatrix} A(S) & B(S) \\ \mathbf{0} & I \end{pmatrix} \tag{15}$$

where $A(S)$ are the transition probabilities between transient states, $B(S)$ are the transition probabilities from transient states to absorbing states, and $I$ is the identity matrix. When clear from context we will drop the dependence of $A, B$ on $S$. Note that $\rho(S)$ has at least one absorbing state (namely $\sigma$). We denote by $\boldsymbol{b}$ the column of $B$ corresponding to state $\sigma$ (i.e., $B$'s rightmost column).

We would like to calculate $f(S)$. By Eq. (6), the probability of reaching $\sigma$ given an initial state $i$ is:

$$c_i(S) = \sum_{t=0}^{\infty} \sum_{j \in [n-r]} \mathbb{P}_S\left[X_t = \sigma | X_{t-1} = j\right] \mathbb{P}_S\left[X_{t-1} = j | X_0 = i\right] = \left(\sum_{t=0}^{\infty} A^t \boldsymbol{b}\right)_i$$

The above series has a closed form solution:

$$\sum_{t=0}^{\infty} A^t = (I - A)^{-1} \quad \Rightarrow \quad \boldsymbol{c} = (I - A)^{-1} \boldsymbol{b}$$

Thus, $\boldsymbol{c}(S)$ is the solution of the set of linear equations, which readily gives us $f(S)$:

$$f(S) = \langle \boldsymbol{\pi}, \boldsymbol{c} \rangle \quad \text{s.t.} \quad (I - A(S))\boldsymbol{c} = \boldsymbol{b}(S) \tag{16}$$

The greedy algorithm can thus be implemented by sequentially considering candidate sets $S$ of increasing size, and for each $z$ calculating $f(S \cup \{z\})$ by solving a set of linear equations (see Algorithm 1). Though parallelizable, this naïve implementation may be costly as it requires solving $O(n^2)$ sets of $n - r$ linear equations, one for every addition of $z$ to $S$. Fast submodular solvers [7] can reduce the number of $f(S)$ evaluations by an order of magnitude. In addition, we now show how a significant speedup in computing $f(S)$ itself can be achieved using certain properties of $f(S)$.

A standard method for solving the set of linear equations $(I - A)\boldsymbol{c} = \boldsymbol{b}$ if to first compute an $LUP$ decomposition for $(I - A)$, namely find lower and upper diagonal matrices $L, U$ and a permutation matrix $P$ such that $LU = P(I - A)$. Then, it suffices to solve $Ly = Pb$ and $Uc = y$. Since $L$ and $U$ are diagonal, solving these equations can be performed efficiently. The costly operation is computing the decomposition in the first place.

Recall that $\rho(S)$ is composed of rows from $Q^+$ corresponding to $S$ and rows from $Q$ corresponding to $[n] \setminus S$. This means that $\rho(S)$ and $\rho(S \cup \{z\})$ differ only in one row, or equivalently, that $\rho(S \cup \{z\})$ can be obtained from $\rho(S)$ by adding a rank-1 matrix. Given an $LUP$ decomposition of $\rho(S)$, we can

efficiently compute $f(S \cup \{z\})$ (and the corresponding decomposition) using efficient rank-1-update techniques such as Bartels-Golub-Reid [17], which are especially efficient for sparse matrices. As a result, it suffices to compute only a *single $LUP$* decomposition once at the beginning, and perform cheap updates at every step. We give an efficient implementation in the supp. material.

# 7   Optimal Tagging

In this section we return to the task of optimal tagging and show how the Markov chain optimization framework described above can be applied. We use a random surfer model, where a browsing user hops between items and tags in a bipartite Markov chain. In its explicit form, our model captures the activity of browsing users whom, when viewing an item, are presented with the item's tags and may choose to click on them (and similarly when viewing tags).

In reality, many systems also include direct links between related items, often in the form of a ranked list of item recommendations. The relatedness of two items is in many cases, at least to some extent, based on their mutual tags. Our model captures this notion of similarity by indirect transitions via tag states. This allows us to encode tags as variables in the objective. Furthermore, adding direct transitions between items is straightforward as our results apply to general Markov chains. Note that in contrast to models for tag recommendation, we do not need to explicitly model users, as our setup defines only one distinct optimization task per item.

In what follows we formalize the above notions. Consider a system of $m$ items $\Omega = \{\omega_1, \ldots, \omega_m\}$ and $n$ tags $T = \{\tau_1, \ldots, \tau_n\}$. Each item $\omega_i$ has a set of tags $T_i \subseteq T$, and each tag $\tau_j$ has a set of items $\Omega_j \subseteq \Omega$. The items and tags constitute the states of a bipartite Markov chain, where users hop between items and tags. Specifically, the transition matrix $\rho$ can have non-zero entries $\rho_{ij}$ and $\rho_{ji}$ for items $\omega_i$ tagged by $\tau_j$. To model the fact that browsing users eventually leave the system, we add a global absorbing state $\varnothing$ and add transition probabilities $\rho_{i\varnothing} = \epsilon_i > 0$ for all items $\omega_i$. For simplicity we assume that $\epsilon_i = \epsilon$ for all $i$, and that $\boldsymbol{\pi}$ can be non-zero only for tag states.

In our setting, when a new item $\sigma$ is uploaded, its owner may choose a set $S \subseteq T$ of at most $k$ tags for $\sigma$. Her goal is to choose $S$ such that the probability of an arbitrary browsing user reaching (or equivalently, being absorbed in) $\sigma$ while browsing the system is maximal. As in the general case, the choice of $S$ affects the transition matrix $\rho(S)$. Denote by $P_{ij}$ the transition probability from item $\omega_i$ to tag $\tau_j$, by $R_{ji}(S)$ the transition probability from $\tau_j$ to $\omega_i$ under $S$, and let $\boldsymbol{r}_j(S) = R_{j\sigma}(S)$. Using Eq. (15), $\rho$ can be written as:

$$\rho(S) = \begin{pmatrix} A & B \\ \mathbf{0} & I_2 \end{pmatrix}, \quad A = \begin{pmatrix} \mathbf{0} & R(S) \\ P & \mathbf{0} \end{pmatrix}, \quad B = \begin{pmatrix} \mathbf{0} & \boldsymbol{r}(S) \\ \mathbf{1} \cdot \epsilon & \mathbf{0} \end{pmatrix}, \quad I_2 = \begin{pmatrix} 1 & 0 \\ 0 & 1 \end{pmatrix}$$

where $\mathbf{0}$ and $\mathbf{1}$ are appropriately sized vectors or matrices. Since we are only interested in selecting tags, we may consider a chain that includes only the tag states, with the item states marginalized out. The transition matrix between tags is given by $\rho^2(S) = R(S)P$. The transition probabilities from tags to $\sigma$ remain $\boldsymbol{r}(S)$. Our objective of maximizing the probability of reaching $\sigma$ under $S$ is then:

$$f(S) = \langle \boldsymbol{\pi}, \boldsymbol{c} \rangle \quad \text{s.t.} \quad (I - R(S)P)\, \boldsymbol{c} = \boldsymbol{r}(S) \tag{17}$$

which is a special case of the general objective presented in Eq. (16), and hence can be optimized efficiently. In the supplementary material we prove that this special case is still NP-hard.

# 8   Experiments

To demonstrate the effectiveness of our approach, we perform experiments on optimal tagging in data collected from Last.fm, Delicious, and Movielens by the HetRec 2011 workshop [3]. The datasets include all items (between 10,197 and 59,226) and tags (between 11,946 and 53,388) reached by crawling a set of about 2,000 users in each system, as well as some metadata.

For each dataset, we first created a bipartite graph of items and tags. Next, we generated 100 different instances of our problem per dataset by expanding each of the 100 highest-degree tags and creating a Markov chain for their items and their tags. We discarded nodes with less than 10 edges.

To create an interesting tag selection setup, for each item in each instance we augmented its true tags with up to 100 similar tags (based on [18]). These served as the set of candidate tags for which

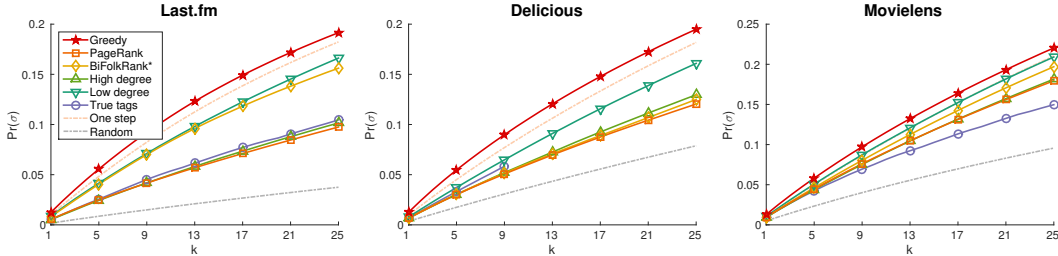

Figure 1: The probability of reaching a focal item $\sigma$ under a budget of $k$ tags for various methods.

transitions to the item were allowed. We focused on items which were ranked first in at least 10 of their 100 candidate tags, giving a total of 18,167 focal items for comparison. For each such item, our task was to choose the $k$ tags which maximize the probability of reaching the focal item.

Transition probabilities from tags to items were set to be proportional to the item weights - number of listens for artists in Last.fm, tag counts for bookmarks in Delicious, and averaged ratings for movies in Movielens. As the datasets do not include explicit weights for tags, we used uniform transition probabilities from items to tags. The initial distribution was set to be uniform over the set of candidate tags, and the transition probability from items to $\varnothing$ was set to $\epsilon = 0.1$.

We compared the performance of our greedy algorithm with several baselines. Random-walk based methods included PageRank and an adaptation[2] of BiFolkRank [10], a state-of-the-art tag recommendation method that operates on item-tag relations. Heuristics included choosing tags with highest and lowest degree, true labels (for relevant $k$-s) sorted by weight, and random. To measure the added value of long random walks, we also display the probability of reaching $\sigma$ in one step.

Results for all three datasets are provided in Fig. 1, which shows the average probability of reaching the focal item for values of $k \in \{1, \ldots, 25\}$. As can be seen, the greedy method clearly outperforms other baselines. Considering paths of all lengths improves results by a considerable 20-30% for $k = 1$, and roughly 5% for $k = 25$. An interesting observation is that the performance of the true tags is rather poor. A plausible explanation for this is that the data we use are taken from collaborative tagging systems, where items can be tagged by any user. In such systems, tags typically play a categorical or hierarchical role, and as such are probably not optimal for promoting item popularity. The supplementary material includes an interesting case analysis.

# 9 Conclusions

In this paper we introduced the problem of optimal tagging, along with the general problem of optimizing probability mass in Markov chains by adding links. We proved that the problem is NP-hard, but can be $(1 - \frac{1}{e})$-approximated due to the submodularity and monotonicity of the objective. Our efficient greedy algorithm can be used in practice for choosing optimal tags or keywords in various domains. Our experimental results show that simple heuristics and PageRank variants underperform our disciplined approach, and naïvely selecting the true tags can be suboptimal.

In our work we assumed access to the transition probabilities between tags and items and vice versa. While the transition probabilities for existing items can be easily estimated by a system's operator, estimating the probabilities from tags to *new* items is non-trivial. This is an interesting problem to pursue. Even so, users do not typically have access to the information required for estimation. Our results suggest that users can simply apply the greedy steps sequentially via trial-and-error [9].

Finally, since our task is of a counterfactual nature, it is hard to draw conclusions from the experiments as to the effectiveness of our method in real settings. It would be interesting to test it in realty, and compare it to strategies used by both lay users and experts. Especially interesting in this context are competitive domains such as ad placements and viral marketing. We leave this for future research.

**Acknowledgments:** This work was supported by the ISF Centers of Excellence grant 2180/15, and by the Intel Collaborative Research Institute for Computational Intelligence (ICRI-CI).

## Footnotes

[1] In an ergodic chain with one absorbing state, all walks reach $\sigma$ w.p. 1, and the problem becomes trivial.

[2]To apply the method to our setting, we used a uniform prior over user-tag relations.

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
