[Supplementary Material]

# Supplementary Material:
# Optimal Tagging with Markov Chain Optimization

**Nir Rosenfeld**
School of Computer Science and Engineering
Hebrew University of Jerusalem
nir.rosenfeld@mail.huji.ac.il

**Amir Globerson**
The Blavatnik School of Computer Science
Tel Aviv University
gamir@post.tau.ac.il

## 1 NP-Hardness of Tag Optimization

In the paper we proved that the general task of maximizing traffic by choosing incoming links is NP-hard. Since our tag optimization formulation is a specific case of the above, its hardness is not immediate. We now prove that tag optimization is indeed NP-hard via a reduction from vertex cover, similar to that of the general case found in Sec. 4 in the paper.

Given a graph $G = (V, E)$, we create a bipartite Markov chain with a state $\tau_i$ for every node $i \in V$ and a state $\omega_{ij}$ for every edge $(i, j) \in E$, and add two absorbing states $\varnothing$ and $\sigma$. We set the transition probabilities in $q$ to be $\frac{1}{d_i}$ from $\tau_i$ to any $\omega_{ij}$, $(1 - \epsilon)$ from $\omega_{ij}$ to $\tau_j$, and $\epsilon$ from $\omega_{ij}$ to $\varnothing$, and continue as in the general case.

## 2 An Efficient Implementation of the Greedy Algorithm

Algorithm 2 displays an efficiently implemented variation of the greedy algorithm proposed in the paper. Its improved runtime stems from two sources. The first is a potentially reduced number of evaluations of $f(S)$. This is achieved by applying well-known submodular maximization methods. The second is an efficient computation of $f(S \cup \{z\})$ given that $f(S)$ has already been computed, using rank-1 $LUP$ updates as described in the paper, and on which we here elaborate.

### 2.1 Reducing the number of evaluations

The literature contains several methods where the submodularity of $f$ is used to achieve a considerable reduction in the number of computations. For instance, the CELF [3] algorithm for submodular maximization maintains a list of lazy evaluations of the marginal gains $f(S \cup \{z\}) - f(S)$, sorted by non-decreasing values. The submodularity of $f$ ensures that marginal gains can only decrease as $S$ grows. Hence, a lazy evaluation serves as an upper bound on the true marginal gain. The algorithm repeatedly updates the marginal gain of the state at the top of the list. If after the update the state maintains its rank, then it is guaranteed to be the correct greedy choice. Otherwise, the list is resorted.

Algorithm 2 optimizes the selection of tags for our objective using CELF-like updates. In our experiments, using CELF resulted in roughly $2n'$ evaluations of $f(S)$ on average for $k = 25$, where $\|\boldsymbol{b}\|_0 = n' < n$. Other popular methods include CELF++ [2] and UBLF [5], of which CELF++ is especially straightforward to implement.

### 2.2 Reducing the cost of each evaluation

A straightforward way to compute $f(S)$ is to solve the set of linear equations in Eq. 16. Nonetheless, the total number of computations can be significantly reduced due to the fact that for any $S$ and $z \notin S$, the matrices $A(S)$ and $A(S \cup \{z\})$ differ only in one row.

---

**Algorithm 2**

---

1: **function** EFFICIENTGREEDYTAGOPT$(Q^+, Q, \boldsymbol{\pi}, k)$
2:     Initialize $S = \emptyset$, $v^0 \leftarrow 0$, and $\forall z \in [n]$   $\delta_z \leftarrow \infty$
3:     Solve $\boldsymbol{c}^0 = \left(I - A(S)\right) \setminus \boldsymbol{b}(S)$ with an $LUP$ solver         $\triangleright$ Using $A, \boldsymbol{b}$ from Eqs. (3), (15)
4:     **for** $i \leftarrow 1$ to $k$ **do**
5:         $\forall z \in [n] \setminus S$   $cur_z \leftarrow$ false
6:         **while** true **do**
7:             $z^* \leftarrow \operatorname{argmax}_{z \in [n] \setminus S} \delta_z$             $\triangleright$ E.g. by using a priority queue
8:             $\boldsymbol{c} \leftarrow$ RANKONEUPDATE$(\boldsymbol{c}^{i-1}, Q^+_{z^*\cdot}, z^*)$    $\triangleright\, \boldsymbol{c} = (I - A(S \cup \{z^*\})) \setminus \boldsymbol{b}(S \cup \{z^*\})$
9:             $v \leftarrow \langle \boldsymbol{\pi}, \boldsymbol{c} \rangle$
10:            **if** $cur_{z^*} =$ true **then**
11:                $S \leftarrow S \cup \{z^*\}$,   $\boldsymbol{c}^i \leftarrow \boldsymbol{c}$,   $v^i \leftarrow v$
12:                **break**
13:            **else**
14:                $\delta_{z^*} \leftarrow v - v^{i-1}$,   $cur_{z^*} \leftarrow$ true         $\triangleright$ Update queue
15:     Return $S$

---

A standard method for solving the set of linear equations $(I - A)\boldsymbol{c} = \boldsymbol{b}$ if to first compute an $LUP$ decomposition for $(I - A)$, namely find lower and upper diagonal matrices $L, U$ and a permutation matrix $P$ such that $LU = P(I - A)$. Then, if suffices to solve $Ly = Pb$ and $Uc = y$. Since $L$ and $U$ are diagonal, solving these equations can be performed efficiently with forward and backward substitution; the costly operation is computing the decomposition in the first place.

Since $A(S)$ and $A(S \cup \{z\})$ differ only in the $z^{\text{th}}$ row, so do $(I - A(S))$ and $(I - A(S \cup \{z\}))$. As we show next, given an $LUP$ decomposition of $(I - A(S))$, a decomposition of $(I - A(S \cup \{z\}))$ can be computed efficiently.

To see why, note that:

$$(LU)_{\Pi_i\cdot} = P_{\Pi_i\cdot}(I - A(S)) = \begin{cases} \boldsymbol{e}_i - Q_{i\cdot} & \text{if } i \neq z \\ \boldsymbol{e}_i - Q^+_{i\cdot} & \text{if } i = z \end{cases}$$

where $\Pi$ is the permutation given by $P$, and $\boldsymbol{e}_i$ is an indicator vector. A similar expression for $S \cup \{z\}$ differs only in the $\Pi_z^{\text{th}}$ row:

$$\begin{aligned} P_{\Pi_z\cdot}(I - A(S \cup \{z\})) &= \boldsymbol{e}_z - Q^+_{z\cdot} \\ &= \boldsymbol{e}_z - Q_{z\cdot} + (Q_{z\cdot} - Q^+_{z\cdot}) \\ &= P_{\Pi_z\cdot}\left[(I - A(S)) + \boldsymbol{e}_z(Q_{z\cdot} - Q^+_{z\cdot})^\top\right] \end{aligned}$$

This offers a new decomposition:

$$L'U = P(I - A(S \cup \{z\})), \quad L' = L + \boldsymbol{e}_z(\boldsymbol{\ell}^\top - L_{\Pi_z\cdot})$$

Where $\boldsymbol{\ell}$ is the solution to $\boldsymbol{\ell}^\top U = Q_{z\cdot}$. Since $L'$ is obtained from $L$ by an addition of a rank-1 matrix, the LUP decomposition of $(I - A(S \cup \{z\}))$ can by computed efficiently using rank-1 update methods such as Bartels-Golub-Reid [4] or Forrest-Tomlin [1]. These methods build on the fact that while $L'$ is not triangular, it can be efficiently transformed into a lower triangular matrix by a small number of row or column swaps.[1] The above methods are especially efficient for sparse matrices (e.g., [4]).

As a result, in both the naïve implementation and in CELF, it suffices to compute a single $LUP$ decomposition for the input at the beginning, and perform cheap updates at every step.

## 3   Experimental Test Case: True Tags vs. Greedy

Figure 1 displays a test-case analysis of the tags selected for a focal item by the greedy algorithm, and the item's true tags ordered by their weight. In this example, the greedy algorithm chose several

Figure 1: A test-case comparison of the tags selected by the greedy algorithm and the item's true tags. Both share some categorical tags, but in reverse order. The greedy algorithm also chose several cross-categorical tags.

categorical tags (such as "groove" and "synth pop") that appear in the set of true tags as well. The order of these tags in the greedy selection is however opposite from that of the true tags, suggesting that being well connected to less popular tags may be more beneficial. The greedy algorithm also selected several cross-categorical tags such as "makes me happy" and "one of my favorites". While probably not very popular on their own, due to their wide degree of connectivity they are likely candidates.

## Footnotes

[1] The efficient LUSOL package for sparse rank-1 $LU$ modifications can be found at `https://web.stanford.edu/group/SOL/software/lusol/`.