[Reviews · NeurIPS 2016]

Reviewer 1

Summary

The article provides a method for assigning tags to items in such a way as to maximize probability of encountering items via random clicking behaviour of users. User follows items and other associated items through the tags attached to the items. Tags with low-number of items boost the probability of random encounters in specific models of user-behaviour. The article is reasonably cleanly written, although not well motivated. The relationship of tags and the actual items is not used in the algorithm, and user satisfaction of encountering certain items is not taken into account. Maximizing hit probability regardless of relevance of those hits seems like trying to game the search process of users in order to expose them to items they do not necessarily want to be exposed to.

Qualitative Assessment

The article is sound, and the markov-chain based assignment algorithm fits the problem. The crux however is the problem formulation - is the problem solved by the proposed method truly a problem? Better motivating the research and motivating the problem would help the reader to appreciate the potential use of the method and its applicability to related areas. The experiments are described briefly, and do not allow the reader to fully recreate the experimental setups. The metric used to evaluate the algorithms suits the proposed method, but doesn't align, in general, with the objectives of those systems - which are to enable users to find the most relevant content for their preferences. How does the proposed method performs in that respect is not explored.

Confidence in this Review

2-Confident (read it all; understood it all reasonably well)


Reviewer 2

Summary

This paper considers the problem of optimal tagging: assigning tags and keywords to online content, in order to maximize traffic. They use a generalization of the random surfer model, where traffic is modeled as a Markov chain. Under this formulation, they show the problem to be NP-hard but non-decreasing submodular and hence easy to approximate by a greedy algorithm.

Qualitative Assessment

The manuscript is well organized and written. The ideas in the paper are good and novel in the context of tagging. The authors are diligent in their discussion of related work, as well as the couple of weaknesses in the paper: i) The fact that the discussed model does not include direct links between items. ii) The fact that the experiments are counterfactual in nature (they do not exactly represent a real system) In both cases the authors provide convincing arguments and suggestions for future work. Overall, this paper was a pleasure to read and I would like to see it accepted.

Confidence in this Review

2-Confident (read it all; understood it all reasonably well)


Reviewer 3

Summary

This paper presents an interesting tagging problem, which clearly has a different goal with tag recommendation studies. To maximize traffic, this paper formulates the optimal tagging problem using a Markov chain, which results in maximizing a certain function over subsets subject to a cardinality constraint. The experimental results shows that the proposed approach outperformed baselines of PageRank, BiFolkRank, High/Low degree, true tags, one step, and random.

Qualitative Assessment

Given 1) this work proposes a new tagging problem, which will have a broad impact on real applications and 2) this problem is well-formulated and solved with an elegant solution in general, I believe this paper deserves publication at NIPS. It would be interested to see some qualitative comparisons between these baseline approaches given several items. Given the numbers shown in Figure 1 are low in general, some real examples are helpful to show the value of the proposed approach.

Confidence in this Review

2-Confident (read it all; understood it all reasonably well)


Reviewer 4

Summary

This paper addresses the problem of tagging an item with a set of K tags such that the probability of a random walk reaching the item is maximized. To this end, the authors formulate this problem as markov chain and determine how the transitions in this such a chain should be modified to increase the traffic to a certain state. The resulting optimization problem is NP-hard but has a 1-1/e greedy approximation due to monotonicity and submodularity properties. The performance of the proposed algorithm is benchmarked on three different datasets.

Qualitative Assessment

This paper is well written and interesting in general, however I am not convinced about its novelty given the state-of-the-art and a weak experimental evaluation further makes it harder to determine the effectiveness of the proposed framework. While this paper provides adequate discussion about the optimization problem, motivation etc., I think it lacks a discussion on how it is different w.r.t: Olsen et. al. A constant-factor approximation algorithm for the link building problem (cited in the paper) Olsen et. al. Maximizing PageRank with New Backlinks (not cited) Olsen et. al. On the approximability of the link building algorithm (not cited) Olsen et. al. An approximation algorithm for the link building problem (not cited) A lot of ideas in this work seem to be borrowed from the aforementioned papers thus limiting the novelty of the proposed approach and formulation. The experimentation setting seems reasonable, however the baselines are rather weak. It would be more convincing to include a baseline from one of the aforementioned papers and see how well it does in comparison with the proposed algorithm.

Confidence in this Review

2-Confident (read it all; understood it all reasonably well)


Reviewer 5

Summary

The current paper is devoted to the problem of maximizing the probability of reaching a given vertex under a Markov random walk on a graph by adding a set of new incoming links from a given set of feasible links under a budget constraint. Authors prove that the problem is NP-hard and propose a greedy algorithm, which is reasonable due to the submodularity of the reaching probability with respect to the chosen set of links. Though the paper is clear and results look correct, the paper does not seem novel enough in the theoretical part and does not seem to have a proved impact for practical tasks. At this stage, I recommend to reject it.

Qualitative Assessment

Optimization of the link structure for PR is not a new topic. Apart from papers mentioned in Related work, there are also those not reviewed, including “PageRank Optimization by Edge Selection” by Csaji et al., “Maximizing PageRank with New Backlinks” by Olsen, “PageRank Optimization in Polynomial Time by Stochastic Shortest Path Reformulation” by Csaji et al. **The novelty** of the study is questionable. The probability of reaching the target state $\sigma$ can be viewed as the state’s stationary probability for the graph, where the added edges are directed to the state $\sigma$ and the matrix of transition probabilities is raised to an appropriate power. This observation does not immediately reduce the problem of the paper to a known task, however, it may partially explain the similarity between the theoretical part and the works of Olsen, where the stationary probability is maximized. In particular, Section 4 resembles the work “Maximizing PageRank with New Backlinks” (not cited in the paper), where M. Olsen considered a reduction of a Markov chain optimization problem to the independent set problem, which is equivalent to the vertex cover problem. Theorems 5.1, 5.3 are reasonable, but very simple and resemble Lemmas 1,2 from [15]. **The practical example** with tags and objects looks a bit questionable, because of the two reasons: 1) To use the proposed solution in practice, one needs first to choose the set of feasible tags for the object. It should be conducted with some objective, and we can assume that it is performed by some ranking algorithm on the graph. Note that the development of such an algorithm is the task of the current paper. Therefore, we can assume in practice that the feasible set coincides with the whole set of tags. Furthermore, this means that objects do not differ in the sense that the properties of the object are not taken into account in the problem formulation. 2) I did not get why maximizing the probability of reaching an item is a well-suited objective for the application. Why is it better than the PageRank value? **Experiments:** I would be appreciated to see a stronger baseline like the following one. Compute, for each tag $\tau$, the probability $P(\tau)$ of reaching $\tau$ for the Markov chain for the initial graph. Choose $k$ tags $\tau$ with the maximal value of $q_{\tau \sigma}^+ P(\tau)$. This heuristic seems to be an equivalent of the one in [15] (“high PageRank values compared to the outdegree”, see Fig. 1). Note that the Low degree baseline outperforms the High Degree one for all datasets according to Figure 1. While the degree of a tag itself is an ambiguous factor, a more complicates heuristic would perform much better. **Other comments:** The presentation could be significantly optimized in terms of space occupied by different parts. In particular, Abstract and Introduction could be shortened significantly. For example, some sentences are redundant: compere “When an item is tagged, positive probabilities are assigned to transitioning from the tag to the item and vice versa” at line 51 and “Intuitively, tagging an item causes probability to flow from the tag to the item, on account of other items with this tag.” at line 54. It is said 3 times that “the optimization problem is NP-hard” and so on. Equation in step 5 of Algorithm 1 is something not standard. The equation for $B$ in line 255 looks wrong. **Questions:** Could you compare the proof in Section 4 with the technique of Olsen? Is the idea of the proof in Section 4 still novel after the work of Olsen? 50: “Our framework can incorporate many web search click models [9, 3].” Which click models from [9] are incorporated in particular? I believe, that paper do not study any Markov chain models. 64: “we argue that maximizing the probability of reaching an item, as opposed to maximizing the proportion of time an infinite random walk spends in the item’s state, is a better suited objective for the applications we consider.” Why? I don’t think so and argue the opposite. **Misspellings:** 46: “a a set” 79: "One the main roles" -> "One of the main roles" 84: “[5].While” Space is missed 111: Eq. 1: $\max_{S\in [n]}$ should be $\max_{S\subset [n]}$ ----- **UPDATE on Author Feedback** - A: Thank you for suggesting the additional baseline - however it actually does appear in our paper (“One Step” in Fig. 1). It implies from the text (l. 284) that One Step baseline is much simpler and definitely different from that I proposed. In fact, you obtain One Step baseline, if you substitute the *initial distribution* $\pi$ instead of the *probabilities $P(\tau)$ of reaching $\tau$* used in the baseline I proposed. I believe, 1) The scoring in my baseline is a better approximation of the true gain, 2) It is still much more efficient than the Greedy algorithm, 3) It is analogous to a baseline from [15]. - A: We agree that it would be practically wise to first select a feasible set of tags. However, we do not see this as a caveat, since finding a set of K potential tags with k << K << n seems like a very reasonable user-initiated preliminary step. It is still not clear, which objectives could prevent a user from choosing K=n while not preventing from choosing K=k. - A: As our model supports arbitrary transition probabilities from tags to items, these can be set according to click models. For instance, in the cascade model, for item i with rank r in tag t we set Pr(t -> i) = (1-p_1)*(1-p_2)*...*(1-p_{r-1})*p_r. We will elaborate on this. Now this is clear. Still, I read [9] and believe this paper is NOT about cascade click models (unlike [3]). Please, be more precise.

Confidence in this Review

2-Confident (read it all; understood it all reasonably well)


Reviewer 6

Summary

The paper considers the problem of choosing a subset of tags for a new coming item such that the probability of a browsing user reaching that item is maximized. It formulates the searching process as a Markov chain and the new item as an absorbing state, so the goal becomes to maximize the probability of the Markov chain being absorbed by the new item. It proves that the problem is NP-hard, but has a (1-1/e)-approximation via a simple greedy algorithm due to mononicity and submodularity.

Qualitative Assessment

The paper proposes a novel optimal tagging method based on absorbing Markov chains, and proves its applicability of simple greedy algorithm. The theory and proofs seem to be sound, and the results are well analyzed. The distinction to the following two closely related works seems quite necessary: 1. [15] in the submission, which is the most related work as mentioned by the authors. The development in this submission seems quite similar to [15], including problem formulation and solution used. 2. Inverting a steady state, wsdm'15, which also optimizes transition probabilities to achieve certain behavior objective. 3. Learning with partially absorbing random walks, nips'12, which introduces a natural extension of absorbing random walks that seems to improve the state of the art in various graph-based learning problems including labeling/tagging on graph.

Confidence in this Review

2-Confident (read it all; understood it all reasonably well)